# Definitions of successful aging among middle-aged Latinas residing in a rural agricultural community

Elizabeth Ambriz[1]*, Camila De Pierola[2], Norma M. Calderon[3], Lucia Calderon[1], Katherine Kogut[2], Julianna Deardorff[2], Jacqueline M. Torres[1]

1 Department of Epidemiology & Biostatistics, University of California San Francisco, San Francisco, California, United States of America, 2 School of Public Health, University of California Berkeley, Berkeley, California, United States of America, 3 Center for Environmental Research and Community Health (CERCH), University of California Berkeley, Berkeley, California, United States of America

* elizabeth.ambriz@ucsf.edu

**Data Availability Statement:** The focus group data hold potentially identifying sensitive information regarding the participants, and it would therefore

## Abstract

### Introduction

Latinos are the fastest growing aging population in the U.S. However, there has been limited attention to conceptualizing successful aging among Latinos, especially those residing in rural communities. Latinos are the largest racial or ethnic group residing in rural underserved communities and rural Latinos experience significant structural barriers to access the conditions they need to age well. The goal of this study is to make unique contributions to the successful aging literature by describing what successful aging means for middle-aged Latinas residing in a rural community.

### Methods

This qualitative paper used inductive thematic content analysis to examine definitions of successful aging among Latina women (n = 40) residing in an underserved agricultural community and entering mid-life (mean = 49 years old; age range 40–64).

### Results

With regards to definitions of successful aging, four themes emerged: 1) Having good health; 2) maintaining an active lifestyle; 3) the wellbeing of one's children; and 4) being independent.

### Discussion

Participants' definitions of successful aging aligned to some extent with existing frameworks, specifically related to health and independence. However, middle-aged Latina participants' unique definitions of successful aging also diverged from existing frameworks, especially around the wellbeing of their children and the importance of work as a way of maintaining an active lifestyle. More research is needed to understand the unique social context and circumstances of middle-aged Latinos residing in rural communities and how

be unethical to make this public and would undermine the minimal risk ethical committee agreement and consent process. Please contact the University of California, San Francisco Institutional Review Board (IRB) to request anonymized focus group data access at CHR@ucsf.edu.

**Funding:** All authors are supported by the National Institute on Aging R01069090. The funders had no role in study design, data collection and analysis, decision to publish, or preparation of the manuscript.

**Competing interests:** The authors have declared that no competing interests exist.

they influence their aging journeys. This can provide important information for the development of culturally sensitive services, interventions, and policies to help Latinos age well.

## Background

Rowe and Kahn's framework for successful aging has been widely adopted in research and defines successful aging as the absence of disease and disability, maintenance of a high degree of physical and cognitive functioning, and meaningful engagement in life [1, 2]. However, this framework has been criticized by the academic community mainly because of its lack of in-depth attention to individuals' and groups' multiple social locations and a narrow conceptualization of "successful aging" that does not capture the diversity in the aging populating itself [3–6]. In particular, there has been limited attention to conceptualizing successful aging among Latinos, the fastest growing aging population in the U.S. [7] and especially Latinos residing in rural communities. As the largest racial or ethnic group residing in rural under-served communities [8], rural Latinos experience more structural barriers to access the conditions they need to age well. In addition, there is limited attention to successful aging among middle-aged adults, although middle age is increasingly understood as a critical window for anticipating and laying the foundation for successful aging through later life [6, 9, 10].

Latinos, especially rural Latinos, in the United States disproportionately lack access to the conditions that support health and its social determinants, including affordable housing, living wages, financial resources, transportation, quality education, and health care access [11]. The health inequities that result from these conditions affect Latinos' ability to age successfully [3]. The health and aging of Latinos may be further compromised by "weathering", or the cumulative wear and tear on the body in response to chronic stressors [12–15]. These structural conditions and their biological impacts put Latinos at elevated risk for high morbidity and mortality attributed to preventable chronic conditions such as diabetes, obesity, high blood pressure, cancer, and mental health conditions [16–18]. These conditions may impact both subjective and objective measures of successful aging including both their physical and cognitive aging patterns.

The few existing studies on successful aging among older Latinos demonstrate that their definitions of successful aging go beyond traditional definitions rooted largely in health and functional outcomes [18–21]. A qualitative study of perceptions on successful aging among a sample of older adults aged 62–88 years in Zapopan, Mexico found that participants' definition of successful aging included acceptance and adaptation to life transitions and health conditions, strong involvement with family and friends, being close to God, the achievement of personal goals, and aging in place [22]. Other qualitative studies about the perceptions of successful aging among Latinos aged 50 or older in the US found similar themes as the study in Mexico; despite having an illness, Latinos focus on maintaining a positive outlook, living in the present, enjoying a sense of community, and finding comfort and meaning in spirituality and family as they age [19–21]. Finally, a quantitative study analyzing cross-sectional data from 9,798 individuals age 60 and above from the Health and Retirement Study (HRS) study found that although older Latino immigrants had lower socioeconomic status as compared to other older adults in the United States, they were the most satisfied with their lives [23]. To our knowledge, Latinos residing in rural communities are not represented in the current aging literature–including literature on successful aging—which is a critical gap give that rural Latinos experience chronic exposure to stressors stemming from structural barriers to access the conditions they need to thrive, and ultimately resulting in poor health outcomes [8, 24].

Furthermore, research on successful aging has prioritized the perspectives of older adults, which is typically thought to be individuals 65 years and over [25], and has not sought to understand the perspectives of middle-aged individuals. Middle-age could be a critical window for understanding what successful aging means for rural Latinos and the development of effective prevention and intervention strategies [6, 9, 10] since Latinos suffer relatively high levels of functional impairment and greater morbidity and mortality associated with preventable chronic diseases such as diabetes [18, 26]. This is particularly important because foreign-born Latina females have a longer total life expectancies at age 50 (36.3 years) compared to White females (33.6 years) and higher prevalence rates of cognitive impairment and dementia, which contributes to fewer number of years spent in a cognitively normal state [27]. Understanding how mid-life Latinas define successful aging gives voice to their unique lived experiences during the critical time of mid-life when people still can redirect and reflect on their own aging journeys. Midlife also represents an excellent opportunity to intervene and design community conditions to support health, prevent disease and disability, and plan for the future [9, 28].

The goal of this study is to make unique contributions to the successful aging literature by describing what successful aging means for middle-aged Latinas residing in a rural community, a group that has not been well represented in research. To do this, we collected qualitative data using focus groups to examine definitions of successful aging among Latina women residing in an underserved agricultural community and entering mid-life (mean = 49 years old).

## Methods

### Participants

Focus group participants were part of the ongoing Center for Health Assessment of Mothers and Children of Salinas (CHAMACOS) Study (UCB IRB current protocol: 2016-08-9072), based in a farmworker community in California. Details of the CHAMACOS study have been published elsewhere [29]. Briefly, participants were originally recruited in 1999/2000 as part of a birth cohort study; an additional group of demographically similar women and children were recruited in 2009 to refresh the sample. While CHAMACOS mothers were in early- to mid-adulthood at the start of the study, they are now primarily in mid-life (mean age for overall study = 47). Follow-up visits with these women and their children have been conducted approximately every two years, with additional quantitative data collected related to the impacts of the COVID-19 pandemic. As of 2021, a total of 594 maternal participants were still part of the CHAMACOS cohort. We included participants in our qualitative study who spoke Spanish (88% of the cohort) and who continued to live in or near the Salinas Valley.

We recruited participants in two ways. First, we recruited participants via the support of the CHAMACOS field staff who distributed recruitment postcards to CHAMACOS mothers attending in person visits as part of the ongoing cohort study. The postcard invited women to contact the first author or share their contact information to be contacted if interested in participating. Second, we sent a recruitment text message to a list of all potentially eligible Spanish speaking CHAMACOS mothers who had participated in any recent aspect of CHAMACOS research (n = 467). Among those participants who requested to be contacted or replied that they were interested in participating (n = 202), the research team followed up via phone call to a subset (n = 70) to schedule focus groups. Of those contacted via phone, a subset (n = 30) were not available to participate, were not reachable, or were not able to make it to the focus group. We recruited participants until saturation was reached and referred the other (n = 128) who were interested in participating to another ongoing qualitative research study led by the first author. All participants in the focus groups (n = 40) were female, had children, spoke

**Table 1. Demographic characteristics of middle-aged Latina women in a qualitative study of definitions of successful aging (N = 40).**

|  | n (%) |
| --- | --- |
| **Ethnicity** |  |
| Mexican | 38 (92.5) |
| Salvadoran | 3 (7.5) |
| **Age** |  |
| 40–49 | 24 (60) |
| 50–59 | 14 (35) |
| 60+ | 2(5) |
| **Employment** |  |
| Unemployed | 12 (30) |
| Agriculture | 22 (55) |
| Non-agriculture | 6 (15) |

Spanish, had an age range of 40–64, and more than half of participants worked in agriculture. Please see Table 1. For demographic characteristics of the sample.

## Data collection

Data were collected from seven focus groups including (n = 40) participants or an average of 5.7 participants per group, at which point saturation was reached. Data collection started in November 2021 and ended in February 2022. The Committee of Human Research of the [REDACTED FOR SUBMISSION] approved the study (21–34842). The overall goal of the focus groups was to examine definitions of successful aging among middle-aged Latina women. The focus groups were organized by age with participants in similar age categories (40s vs. 50s to early 60s) participating in a focus group together. The focus groups were organized by age to elucidate differences in definitions of successful aging by age. Focus groups were conducted in Spanish by the first author, who is a bilingual and bicultural Latina researcher, were designed to last about 120 minutes, and were audio recorded and professionally transcribed. Focus groups were held in person in the CHAMACOS field office or at a local park conveniently located for participants. Participants received a $75 gift card for their time. Focus groups began with the facilitator obtaining informed written consent. The facilitator then began posing questions from the focus group guide starting with probing participants' own definitions of successful aging. Then, participants were asked to describe someone who they thought was aging successfully. Next, the facilitator asked if they thought they were aging successfully and what barriers they experience to age successfully. Finally, participants were asked what would make them satisfied in old age, and their worries about aging (S1 Appendix). The facilitator took field notes during and after the focus groups.

## Data analysis

Transcripts of focus groups were entered into MAXQDA software. The first author and a research assistant analyzed the transcripts with inductive thematic content analysis (Angel, 2009) [3]. Transcripts were analyzed in Spanish line by line [30] to examine definitions of successful aging among middle-aged Latina women. First, we focused on identifying themes and subthemes [31]. Next, we compared themes from each focus group to assess whether they corroborated, negated, or expanded one another, which adds depth to the analysis [32]. To validate the data, we used collaborative coding, the first author and a research assistant independently coded each transcript using an iterative process [33]. First, we individually read

transcripts to generate an initial list of key concepts and group repeated concepts into patterns or themes. Then we met, discussed, and came to consensus about the code structure before re-reading additional transcripts searching for these or other new themes. We analyzed focus group data in an iterative process reviewing and refining themes until no new concepts emerged from subsequent focus groups and resulting themes are deemed "saturated" by the analytic team [34] This study is reported as per the Consolidated Framework for Reporting Qualitative Research (COREQ) guideline (S1 Checklist) [35]. Finally, the first author translated quotes used to summarize each theme and subtheme.

## Results

With regards to definitions of successful aging, four themes and various subthemes emerged under each theme: 1) Having good health and accepting changes in physical body; 2) maintaining an active lifestyle; 3) the wellbeing of one's children; and 4) being independent.

### Having good health

Participants across focus groups defined successful aging as having "good health," while the theme of accepting the physical changes in their bodies with age as key to being happy was more common among the 50s-early 60s focus groups. One participant summarized her definition of successful aging as, "Yes, I'm aging successfully because one has good health and, when I go to the doctor, well, everything is good with my health" (Participant in the 40s focus group).

Participants expressed that they value their health, even though some said that they are currently not healthy and are afraid of developing chronic conditions such Alzheimer's dementia because they are already forgetting things. One said, "I'd like to have health, have health like others are saying here, have health even though I don't, I don't because lately I've been very sick from depression, I've suffered many years with depression, and I'm afraid that with time, because sometimes I forget things, that with time, I'll forget everything completely. Yes, I'm afraid and I think, I don't think it's because of age, because I'm only 53, but I'm afraid that with the passing of time I won't remember anything" (Participant in 50s-early 60s focus group).

### Being happy despite having an illness

Some participants shared that they are happy, despite having an illness, because they accept and acknowledge the physical changes in their bodies with age. This was illustrated by a participant in the 50s-early 60s focus group when they shared, "Well, I feel well even though I have diabetes. I feel happy with my white hair, I've always been overweight. . . when I was younger, I was thin, but I continued to gain weight once married, but I am happy like this with my husband and my family."

### Accepting changes in physical body

Additionally, physical changes in the body were discussed as part of the "natural" aging process and acceptance as key to being happy and satisfied with one's aging. One participant summarized this as, "I am [laughs] I am aging successfully, even though I sometimes ask myself why I'm overweight? I've always been thin but after 40 I've gained some pounds despite taking care of my diet. I tell myself, I'm going to accept myself, and I accept that when one is older our metabolism is slower and is not going to be the same as when one is 20. I say I am going to

accept it and be happy about how I'm aging, at my stage, I'm happy, at my age that I have I'm happy, I don't feel old" (Participant in the 50s to early 60s focus group).

Participants consider health and accepting the physical changes to their bodies with age as essential for successful aging.

### Maintaining an active lifestyle

For participants, successful aging also meant continuing one's active lifestyle whether it be through work, household demands, walks, and caring for their children.

### Exercise and everyday activities

This theme was agreed upon by the women and summarized when one added, "For me aging successfully would be aging healthily, that I don't have any disease, that I continue to have the same energy to continue doing everyday tasks, working, and taking care of my children and my home" (Participant in the 40s focus group). In this group, an active lifestyle was deemed as essential to maintain health and therefore successful aging of the mind and body. This was summarized by a participant as follows, "I am. I feel like I'm aging satisfactorily because I'm healthy. I can walk. I can come and go. Do things for myself. I can still run after my children, a bit slowly, but I can reach them" (Participant in the 40s focus group). For this group, being active through exercise and everyday activities was essential to support health and successful aging.

### Work

Interestingly, while exercise was discussed through traditional methods such as walking, for this group it also encompassed the physical demands of work. One participant supported this idea when stating, "I want to at least work because I don't like to exercise, one job and another, but don't stop because I don't want to stay and be shut in more than I already I am." (Participant in the 50s-early 60s focus group).

Participants described others who they thought were aging successfully as being active in their everyday lives and continuing to work. This is shown through an example given by one of the participants when discussing their own father, "he still comes, he does not stop working here [in US], or there [Mexico]. We tell him, "don't work anymore," "don't go there anymore," "retire already," and he says, "Ay, my daughter, I can still do it" (Participant in 40s focus group).

Another participant described her uncle as an example of someone aging successfully: "he is now with a hunched back, but he is still helping my cousin to clean his yards, he goes out and says he's looking for a girlfriend [laughs]. They had Banda music for his 100th birthday two weeks ago and he was dancing with his granddaughters, it was beautiful because he already is 100 years" (Participant in 50s-early 60s focus group).

Overall, participants hoped to maintain their health and wellbeing into old age and continue their active way of living for a satisfying aging process.

### The wellbeing of their children

The overwhelming majority (96%) of women in the study defined successful aging in terms of their children. To them, aging successfully means that their children are self-sufficient, independent, have a good job and financial stability to live well, and practice family values. Their children played a significant role when determining their own successful aging, ultimately utilizing their children's success as a measurement for themselves.

### Self-sufficiency

Of all factors discussed in the focus groups, their children's self-sufficiency to live independently was most important. The women felt that preparing their children to live independently from them and taking advantage of the opportunities in the U.S. indicated their success. The push for their children to follow their goals and support themselves was described as, "Like I tell you, my phase of raising, caring, and educating my children is over. Now it is up to them what they want to achieve, but I've helped them as much as I could, my children are all grown-ups, and thanks to God they can support themselves" (Participant in 50s-early 60s focus group).

### Live well

While children's higher education was important to the mothers, many understood that it was beyond the interest of some of their children and therefore prioritized financial stability through employment. The mothers were flexible in terms of their expectations for their children, often citing that being good people and building their career based on their interests was a proud accomplishment. One woman summarizes the importance of her children's independence for her own aging, "I want my children to be fulfilled, yes. That they have their job, maybe someone will not like school, but if I see they live well, for me that would be aging with dignity for myself, because I will be at peace seeing that my children are well. And then I could focus on myself, my husband, and that's it" (Participant in 40s focus group).

### Practice family values

Practicing family values or being good people was described as not being involved in gangs or drug use and, and instead striving to achieve their goals. This sentiment was summarized by a woman, "As a person, as a mother, you feel fulfilled seeing that your children didn't get into trouble. That they're doing something good for themselves. Because I tell them, maybe mom and dad cannot leave you an inheritance, but the only inheritance you will have, I tell them, is going to be your education. What do you want to achieve?" (Participant in 40s focus group).

In all, the women in this study prioritized their children and their wellness, both economically and health-wise, as determinants of their success. Raising their children is seen as an accomplishment and milestone for the mothers, ultimately defining their progression through life.

### Independence

Independence was a key component of successful aging for all participants in the study. When imagining themselves as older, many participants explained the sadness and worry of not being able to take care of themselves. This lack of self-sufficiency and capability was viewed as unsuccessful aging because they would depend on others for basic needs to continue their lifestyle. One participant describes this dependence as, "If you reach a certain age is like becoming a child again because you'll need a diaper, that someone feeds you in the mouth, and that they help you sit. Then, yes, that would be very difficult" (Participant in 40s focus group).

### Preparing now to be independent in old age

With independence, conversation also diverged to feelings of freedom and ability to do what they wanted without external pressure. Additionally, they shared that one could plan to be independent in old age and saw what they were doing today linked to their future independence. "Liberty is, being worry-free, I can go out to walk, sleep, visit someone, I don't have

anything to pressure me. But it has a lot to do with what I'm doing today to prepare, right?" (Participant in 50s-early 60s focus group).

## Not depending on their children

Additionally, most of the mothers added that independence would be exemplified through not depending on their children as they prepared for later life. On the one hand, mothers believed that their children were not obligated to care for them in old age, therefore withholding that expectation. On the other hand, they also revealed that their children's independence and ability to live their own life raised concerns in terms of their motivation to care for them. For this reason, mothers spoke about not relying on their children because of instances where children focused on their own lives and put their parents aside. Interestingly, some mothers also added that they did not want to be a burden on their children and their families, ultimately deciding that being independent was best for all parties to avoid conflict. This concern was illustrated when one participant stated, "For me, like the fellow friend said, I would not like to reach an age when I have to depend on anyone, because when one becomes dependent on children, I have seen cases when the siblings fight because no one wants to care for their parents" (Participant in 40s focus group).

## Maintaining family unity and relationships

Family unity and cohesion were noted to be of importance among participants as they considered the uncertainty of their future and need for independent preparedness. While there were mixed sentiments regarding the caring for future grandchildren, the women agreed they wanted future interactions with their children and grandchildren. One participant highlights this goal when stating, "I would like that my son gets married and gives me grandchildren because I can still help take care of them" (Participant in 40s focus group). However, some participants shared that they did not want to care for their grandchildren full time because they had already taken care of their own children and now wanted to focus on themselves. This was illustrated by one participant as, "I visualize myself with my husband [laughs]. Just the two of us, maybe with our grandchildren sometimes, but not taking care of them all the time. I imagine myself traveling, going to Mexico in seasons. That's all" (Participant in the 40s focus group).

Above all, the women wished to age autonomously due to their familial concerns and ability to be self-sufficient.

Table 2. Summary of themes and subthemes.

## Discussion

Middle aged Latinos residing in rural communities have been severely underrepresented in the successful aging literature. Latinos suffer from high morbidity and mortality due to preventive chronic conditions such as diabetes, high blood pressure, obesity, and cancer–and middle age is a crucial window of opportunity to prevent and manage these conditions. Therefore, it is important to capture Latino perspectives on successful aging in mid-life as this is a critical period during which to develop prevention and intervention strategies for later life.

Participants' definitions of successful aging align to some extent with existing frameworks developed primarily among white participants over 60 years old [6, 9], specifically related to health and independence. They value health and share that accepting the physical changes in their bodies as part of a natural process is key to being happy. This is similar to a qualitative study of 30 Korean, Vietnamese, and Latino older adults, which found that Latinos participants were more likely than their Korean and Vietnamese counterparts to have a positive

**Table 2. Themes/sub themes.**

| | Exemplar Quotes |
|---|---|
| **Having good health** | |
| Being happy despite having an illness | "Well, I feel well even though I have diabetes. I feel happy with my white hair, I've always been overweight. . . when I was younger, I was thin, but I continued to gain weight once married, but I am happy like this with my husband and my family" (participant in the 50s-early 60s focus group). |
| Accepting changes in physical body | "I am [laughs] I am aging successfully, even though I sometimes ask myself why I'm overweight? I've always been thin but after 40 I've gained some pounds despite taking care of my diet. I tell myself, I'm going to accept myself, and I accept that when one is older our metabolism is slower and is not going to be the same as when one is 20. I say I am going to accept it and be happy about how I'm aging, at my stage, I'm happy, at my age that I have I'm happy, I don't feel old" (Participant in the 50s to early 60s focus group). |
| **Maintaining an active lifestyle** | |
| Exercise and everyday activities | "I am. I feel like I'm aging satisfactorily because I'm healthy. I can walk. I can come and go. Do things for myself. I can still run after my children, a bit slowly, but I can reach them" (Participant in the 40s focus group). |
| Work | "He still comes, he does not stop working here [in US], or there [Mexico]. We tell him, "don't work anymore," "don't go there anymore," "retire already," and he says, "Ay, my daughter, I can still do it" (Participant in 40s focus group). |
| **The wellbeing of one's children** | |
| Self-sufficiency | "Like I tell you, my phase of raising, caring, and educating my children is over. Now it is up to them what they want to achieve, but I've helped them as much as I could, my children are all grownups, and thanks to God they can support themselves" (Participant in 50s-early 60s focus group). |
| Live well | "I want my children to be fulfilled, yes. That they have their job, maybe someone will not like school, but if I see they live well, for me that would be aging with dignity for myself, because I will be at peace seeing that my children are well. And then I could focus on myself, my husband, and that's it" (Participant in 40s focus group). |
| Practice family values | "As a person, as a mother, you feel fulfilled seeing that your children didn't get into trouble. That they're doing something good for themselves. Because I tell them, maybe mom and dad cannot leave you an inheritance, but the only inheritance you will have, I tell them, is going to be your education. What do you want to achieve?" (Participant in 40s focus group). |
| **Independence** | |
| Preparing now to be independent in old age | "Liberty is, being worry-free, I can go out to walk, sleep, visit someone, I don't have anything to pressure me. But it has a lot to do with what I'm doing today to prepare, right?" (Participant in 50s-early 60s focus group). |
| Not depending on their children | "For me, like the fellow friend said, I would not like to reach an age when I have to depend on anyone, because when one becomes dependent on children, I have seen cases when the siblings fight because no one wants to care for their parents" (Participant in 40s focus group). |
| Maintaining family unity and relationships | "I would like that my son gets married and gives me grandchildren because I can still help take care of them" (Participant in 40s focus group) |

outlook and being happy despite having an illness [21]. This study found that while participants across focus groups value their health as key to aging successfully, older participants in the 50s- early 60s focus groups also defined successful aging as accepting the physical changes in their bodies with age as key to being happy. Participants' independence or ability to maintain an active lifestyle without depending on their children or other people to do their activities of daily living is of extreme importance. Many shared that while they are still currently independent, they are fearful of the future if they cannot do many essential things without help. It was important for this group to not rely on their children to provide practical or financial help but that they maintain relationships with and proximity to children and grandchildren. Participants placed more emphasis on anticipating a desire for emotional support from their

children than on other forms of assistance. This finding is similar to a qualitative study on positive aging among 101 Latino adults ages 60 and older, which found that Latino older adults value self-sufficiency while at the same time want to maintain strong relationships and connections with their children and extended networks [36].

Middle-aged Latina participants' unique definitions of successful aging also align with prior findings including Chinese immigrants to a Western country, which found that participants saw their children's success and wellbeing as essential to their own sense of aging well [37]. This group defined their own successful aging in terms of the success and wellbeing of their children in many aspects, including having a good job, accomplishing their goals, being good people and practicing family values. They reiterated that they would feel fulfilled and accomplished if their children were doing well. The strong focus on their children might be because all participants in the study have children, and the importance of familism as a cultural value creating a sense of obligation to take care of one's family by providing the necessary emotional and instrumental social support when needed [38]. This finding reinforces previous findings including Chinese immigrants about the role of children on parental wellbeing and their own definitions of successful aging. This is also consistent with findings from other research on the interconnectedness between the wellbeing of aging parents and their children. For example, a quantitative study including 633 middle aged adults found that having even one child suffering from psychosocial problems was associated with poor parental wellbeing [39].

However, one aspect where our results diverge from existing successful aging frameworks is that in this group, maintaining an active lifestyle through work, household chores, and exercise was deemed as essential to maintain health and therefore successful aging of the mind and body. Interestingly, while exercise was discussed through traditional methods such as walking, for this group it also encompassed the physical demands of work. Participants also described others, mainly parents and family members, who they thought were aging successfully as being active and continuing to work until old age. This group also hoped to continue to work until old age in part because they wanted to continue an active lifestyle and in part because they want to be financially independent in old age and see the need to continue working after 65 to be able to accomplish this goal. This may be because Latino and Black workers have lower participation and contributions in employer sponsored retirement plans than do white workers, one of the factors contributing to them reaching the later years of life with substantially fewer economic resources [40]. Furthermore, participants in this study are employed mostly in low-wage jobs in the agriculture or service sectors, and struggle to afford the high cost of living in the Salinas Valley [41].

Newer successful aging frameworks include a consideration of the social context and social determinants of health. This is crucial because rural Latino neighborhoods in the United States possess limited access to health care, internet, transportation, and recreation infrastructure, which negatively impacts health outcomes and behaviors [11, 24]. The socioeconomic position for Latinos matters because overall, they have fewer resources to deal with preventable chronic conditions since they are overrepresented in low-wage jobs with no benefits such as health insurance. Health insurance instability has been associated with worse health outcomes including poor diabetes control among Latinos [26]. Insights from this study can help us better consider how to tailor interventions and preventive models for Latinas that center the wellbeing of the whole family, especially of children, as inseparable from the well-being and successful aging of Latina mothers.

It is important to examine successful aging among this group through a biopsychosocial perspective, which helps us understand the development of illness and health through the complex interaction of biological factors (genetic, biochemical, etc.), psychological factors (mood, personality, behaviour, etc.) and social factors (cultural, familial, socioeconomic, etc.)

[42]. For participants in this study, accepting physical changes in their body, maintaining an active lifestyle through work, and the wellbeing of their children was critical for their own health and successful aging. Findings from this research highlight the interconnections between individual and group perceptions of successful aging and the underlying biology of health and disease. Furthermore, these findings align with existing research that shows a two-way relation between physical health and subjective wellbeing; poor health leads to reduced subjective wellbeing, while high subjective wellbeing can reduce physical health impairments [43]. Furthermore, a recent investigation from the Multi-Ethnic Study of Atherosclerosis (MESA) comprising 1036 participants of Black, Chinese-American, Hispanic and White origin showed that it is not race/ethnicity but cardiovascular risk factors such as age, smoking, hypertension, diabetes and socio-economic status, that determine the outcome on cognitive decline [44]. More research is needed to understand the complex interactions of biopsychosocial factors and their influence on Latino's aging journeys in order to develop tailored prevention and intervention strategies for latter life.

## Strengths and limitations

This study provides unique contributions to the successful aging literature because it includes a group that, to our knowledge, has not been included in successful aging studies, middle-aged Latina women residing in a rural agricultural community. Additionally, the study provides unique perspectives from a sample including more than half of participants who worked in agriculture across seven focus groups.

Although the data saturation was considered to have been achieved with the current participants, a larger number of participants from a wider range of Latino backgrounds residing in rural communities might have fully captured the perspectives of this diverse group. Furthermore, because we recruited participants from a long-term cohort study, we might have only attracted female participants who have been exposed to health-related messages, through educational events, informative newsletters, and other community events tailored to this cohort of women, some of whom have participated in the study for more than twenty years.

## Conclusion

Participants' definitions of successful aging align to some extent with existing frameworks, specifically related to health and independence. However, middle-aged Latina participants' unique definitions of successful aging also diverged from existing frameworks, especially around the wellbeing of their children and the importance of work as a way of maintaining an active lifestyle. More research is needed to understand gender differences in similar research including mixed gender focus groups and the unique social context and circumstances of middle-aged Latinos and their families residing in rural communities and how it influences their aging journeys. Centering on the experiences of the whole family and taking a life course perspective on successful aging can provide important information for the development of culturally sensitive services, interventions, and policies to help Latinos age well. Middle age also represents an opportunity for practitioners and policymakers to support the successful aging of Latinos as they define it by creating the community conditions to support health, prevent disease and disability, and plan for the future.

## Supporting information

**S1 Checklist. COREQ (COnsolidated criteria for REporting Qualitative research) checklist.**
(PDF)

**S1 Appendix. Focus group guide.**
(DOCX)

## Acknowledgments

We thank the participants who have helped us make this research possible. We are grateful to the CHAMACOS program and its staff for their assistance in recruiting focus groups participants and planning the focus groups.

## Author Contributions

**Conceptualization:** Elizabeth Ambriz, Jacqueline M. Torres.

**Data curation:** Elizabeth Ambriz, Julianna Deardorff, Jacqueline M. Torres.

**Formal analysis:** Elizabeth Ambriz, Camila De Pierola.

**Funding acquisition:** Elizabeth Ambriz, Jacqueline M. Torres.

**Investigation:** Elizabeth Ambriz.

**Methodology:** Elizabeth Ambriz, Katherine Kogut.

**Project administration:** Katherine Kogut.

**Resources:** Norma M. Calderon, Lucia Calderon, Jacqueline M. Torres.

**Software:** Lucia Calderon.

**Supervision:** Jacqueline M. Torres.

**Validation:** Camila De Pierola, Norma M. Calderon.

**Writing – original draft:** Elizabeth Ambriz.

**Writing – review & editing:** Camila De Pierola, Lucia Calderon, Katherine Kogut, Julianna Deardorff, Jacqueline M. Torres.

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
