## [Decision Letter · Decision Letter 0]

19 Dec 2022

PONE-D-22-28508Definitions of Successful Aging among Middle-Aged Latinas residing in a Rural Agricultural CommunityPLOS ONE

Dear Dr. Ambriz,

Thank you for submitting your manuscript to PLOS ONE. After careful consideration, we feel that it has merit but does not fully meet PLOS ONE’s publication criteria as it currently stands. Therefore, we invite you to submit a revised version of the manuscript that addresses the points raised during the review process.

We look forward to receiving your revised manuscript.

Kind regards,

Callam Davidson

Editorial Office

PLOS ONE

Journal Requirements:

2. We note you have included a table to which you do not refer in the text of your manuscript. Please ensure that you refer to Table 1 in your text; if accepted, production will need this reference to link the reader to the Table

“All authors are supported by the National Institute on Aging R01069090.”

6. http://www.equator-network.org/reporting-guidelines/strobe/ Please ensure that the study is reported according to the STROBE guideline, and include the completed STROBE checklist as Supporting Information. Please add the following statement, or similar, to the Methods: \\"This study is reported as per the Strengthening the Reporting of Observational Studies in Epidemiology (STROBE) guideline (S1 Checklist).\\"\\n\\nThe STROBE guideline can be found here: http://www.equator-network.org/reporting-guidelines/strobe/\\n\\nWhen completing the checklist, please use section and paragraph numbers, rather than page numbers."}" style="color: rgb(0, 0, 0); font-size: 10pt; font-family: Arial;"> Please ensure that the study is reported according to the COREQ guideline, and include the completed COREQ checklist as Supporting Information. Please add the following statement, or similar, to the Methods: "This study is reported as per the Consolidated criteria for reporting qualitative research http://www.equator-network.org/reporting-guidelines/strobe/"" }�{"1":451}" data-sheets-textstyleruns="{"1":0}�{"1":392,"2":{"2":{"1":2,"2":1136076},"9":1}}�{"1":451}" data-sheets-userformat="{"2":1049345,"3":{"1":0},"11":4,"12":0,"23":1}" data-sheets-value="{"1":2,"2":"Please ensure that the study is reported according to the STROBE guideline, and include the completed STROBE checklist as Supporting Information. Please add the following statement, or similar, to the Methods: \\"This study is reported as per the Strengthening the Reporting of Observational Studies in Epidemiology (STROBE) guideline (S1 Checklist).\\"\\n\\nThe STROBE guideline can be found here: http://www.equator-network.org/reporting-guidelines/strobe/\\n\\nWhen completing the checklist, please use section and paragraph numbers, rather than page numbers."}" style="color: rgb(0, 0, 0); font-size: 10pt; font-family: Arial;">  (COREQ) guideline (S1 Checklist)."

The COREQ guideline can be found here: https://www.equator-network.org/reporting-guidelines/coreq/ 

http://www.equator-network.org/reporting-guidelines/strobe/"}�{"1":451}" data-sheets-textstyleruns="{"1":0}�{"1":392,"2":{"2":{"1":2,"2":1136076},"9":1}}�{"1":451}" data-sheets-userformat="{"2":1049345,"3":{"1":0},"11":4,"12":0,"23":1}" data-sheets-value="{"1":2,"2":"Please ensure that the study is reported according to the STROBE guideline, and include the completed STROBE checklist as Supporting Information. Please add the following statement, or similar, to the Methods: \\"This study is reported as per the Strengthening the Reporting of Observational Studies in Epidemiology (STROBE) guideline (S1 Checklist).\\"\\n\\nThe STROBE guideline can be found here: http://www.equator-network.org/reporting-guidelines/strobe/\\n\\nWhen completing the checklist, please use section and paragraph numbers, rather than page numbers."}" style="color: rgb(0, 0, 0); font-size: 10pt; font-family: Arial;"> When completing the checklist, please use section and paragraph numbers, rather than page numbers.

Additional Editor Comments:

The Data Availability Statement (DAS) requires revision. If the data are freely or publicly available, please include the DOI or accession number).

Reviewers' comments:

Reviewer's Responses to Questions

**Comments to the Author**

1. Is the manuscript technically sound, and do the data support the conclusions?

Reviewer #1: Yes

Reviewer #2: Yes

Reviewer #3: Yes

2. Has the statistical analysis been performed appropriately and rigorously? 

Reviewer #1: N/A

Reviewer #2: N/A

Reviewer #3: N/A

3. Have the authors made all data underlying the findings in their manuscript fully available?

Reviewer #1: Yes

Reviewer #2: No

Reviewer #3: Yes

4. Is the manuscript presented in an intelligible fashion and written in standard English?

Reviewer #1: Yes

Reviewer #2: Yes

Reviewer #3: Yes

5. Review Comments to the Author

Reviewer #1: I enjoyed reading this well-written manuscript, and found the results to be interesting and insightful. The study methods sound appropriate, and the findings provide new insights into Latino perspectives on successful ageing. I would very much like to see this work published, although I do think that there are a few issues that the authors should consider before the manuscript is ready for publication. I outline these below.

Main points

1. Insufficient attention is currently paid to the fact that the study only includes female participants. At the very least, this should be explained and justified in the introduction section. It is also something that could be worth exploring in the discussion section – especially when considering how the current findings differ to those of previous work in Latino populations e.g. what proportion of previous samples were female, and could the smaller proportions of female participants (or, indeed, the method of data collection e.g. if mixed gender focus groups were used) account for the differences in key themes that emerged? Based on these reflections, do you feel that gender differences should be explored further in similar research?

2. Your description of the analytical procedure is a little unclear, and lacks important detail. For instance:

a. What kind of coding did you use (and might you consider adding a table with some examples of the initial codes that you applied to some of the quotes from each theme?)?

b. What do you mean by ‘narratives’ and how do these differ from your themes?

c. More detail is needed to explain how you did your collaborative coding, and how you used the Consolidated Framework.

3. There is a fair bit of overlap between your first two themes, in that Theme 1 (‘Having Good Health and Accepting Changes in the Physical Body’) includes details of participants wanting to maintain active lifestyles, and to continue to take part in valued activities, and Theme 2 (‘Maintaining an Active Lifestyle’) then ends by mentioning participants’ desires to maintain their health and wellbeing. Whilst these two themes are logically related to one another, I felt that the content and narratives could be better organised so that there is a clearer distinction between the focus of each.

4. Whilst the introduction and discussion do a great job of considering what is known about Latino peoples’ perspectives on successful ageing, I think that the paper could be much improved by also considering how these findings compare to findings from other cultures and groups. For instance, which of the findings seem to be somewhat universal across cultures/groups, and which are more specific to Latinos, or to other groups with shared characteristics? For instance, work in this area conducted with Chinese immigrants to a Western country by Teh et al (2020: DOI: 10.1111/ajag.12677) found that participants saw their offspring’s success and wellbeing as integral to their own sense of ageing well. The work also highlights a similar tension between participants wanting their children to be independent, whilst also feeling concern at them not being there to take care of them as they age. It would therefore be useful to situate the current findings within this broader literature base so that such similarities and differences are highlighted.

5. In relation to the previous point, your argument in the discussion about the ‘unique contribution’ of ‘the important role of children’ that this work makes to ‘successful ageing frameworks’ is not accurate, as these points have already been made by authors such as Teh and colleagues.

Minor points

6. There are a couple of missing bits of text in the version of the manuscript that I saw. In particular:

a. The introduction paragraph of the abstract contains a question mark in place of a word (‘..rural Latinos experience more structural barriers than ? to access the conditions…’)

b. The penultimate sentence of the results section ends with empty quotation marks (‘This was illustrated by one participant as ‘’ ‘).

7. It would be useful to state the overall age range of the participants, rather than just the mean age, in the abstract, introduction, and methods section.

8. The first sentence of the second paragraph of the Background section (starting: ‘Latinos, especially rural Latinos, in the United States…’) should be supported with a citation.

9. It would be useful to know in the methods section how and why the particular subset of 70 responders were chosen from the 202 that contacted the research team.

10. The final couple of sentences of text in the ‘participants section’ (that describe the size of the focus groups, the dates of data collection, and the means of ethical approval) would seem better placed in the ‘data collection’ section.

11. I would suggest providing a more complete description of the topic guide in the main manuscript, as many readers will download the paper without the appendix, and so will not have access to this when reading the paper.

12. Could you explain why you chose to organise focus groups by age?

13. Some of the information in the demographics table is not useful. In particular, as you explain elsewhere that all of the participants were female, and that they all spoke Spanish, it doesn’t seem necessary to report the %s of these characteristics. I also felt that these demographic details of your sample might be better placed in your participants section, than your results, as they are descriptions of your sample rather than ‘findings’.

14. In the discussion section you mention that all of the participants had children. If this information was formally collected as part of the study, then it would be useful to report this when describing the characteristics of the sample.

15. It is not clear what you mean in the ‘strengths and limitations section by your sample being ‘robust’.

16. If word count is an issue, then I think that much of the first paragraph of the discussion could be omitted, as it repeats many points from the introduction. Other parts of the discussion also reiterate parts of the results section that are not needed. Some of your quotes could also be cut down.

Reviewer #2: This is a well-written paper on successful aging in Latinas in a rural community. This paper contributes to the literature by providing the perspectives of a unique population. I have some minor comments below for the authors to consider in revising their manuscript.

Introduction: The authors build a strong case for investigating successful aging in middle-aged Latinos. However, the introduction is missing some context as it pertains to women specifically, and related to the life expectancy of Latinas. These pieces are need to provide a rationale for the sample, and to provide context to the results.

Line 160: How did the researchers decide which participants to follow-up with? Why didn’t they follow-up with all participants? Some explanation is required here to provide more understanding of bias in the sample selection.

Line 163: How was it established that saturation was reached?

Table 1: Since only women were recruited, gender can be removed from the table. Also, did the authors ask about gender, or are they using biological sex?

Results: Since the focus groups were done in age-groups, can the authors address age differences in the findings? This comparison would help elucidate some of the issues or gaps that were identified in the introduction of the paper.

Results: Quotes should be pulled out from the paragraphs to improve readability and to highlight them.

Results: The high-level themes appear to have several sub-themes that are not clearly addressed. For example, it appears that weight came up on more than on occasion. Is there a sub-theme related to weight that might need to be highlighted? Similarly, some of the quotes suggest that perhaps the women had low self-esteem or were depressed. But perhaps there is a cultural component pertaining to humility that is being captured in these quotes instead. It would be nice to see a more thorough breakdown of the data. A table or figure could clearly provide the themes and sub-themes. This would also help with highlighting the novelty in the discussion as well i.e. cultural differences, age differences, etc.

Introduction and Discussion: Given that the age of the sample was much younger than samples who are typically included in studies regarding successful aging, it is important for the authors to highlight the sample of the age in the studies being used for comparison. Are there other studies that have asked about successful aging in middle-aged adults that can be used for direct comparison? It is also worth addressing the fact that the probing questions used across studies can vary significantly. For example, the current study included a question asking for examples of people who were aging successfully. Other studies may not have asked this directly, and therefore it may not have arisen as a theme. Comparison to previous literature needs to be put in the context of the age of the sample and the probing questions used.

Line 366: Did all of the women in the sample have children? If such a dimension is added to a framework, will it be inclusive of such women? I’m not sure I agree with the suggestion being made in this line.

Discussion: The rural aspect of the sample did not appear to come through in the discussion, other than in the section on physical demands of work. This is a novel aspect of the work, and might need to be better highlighted in the discussion.

Line 386: typo

Line 57: Typo

Reviewer #3: The authors studied what successful aging means among 40 middle-aged Latinas who participated in focus groups conducted within the context of the CHAMACOS Study in the Salinas Valley of California in 2021-2022.

In the Introduction, the authors provided an adequate summary of the state of the field of successful aging research based on relevant published work, highlighted gaps in knowledge about Latino populations, and articulated the rationale for their work. In the Methods, the authors provided adequate description of participant recruitment, conduct of focus groups, and thematic content analysis of the qualitative data. As reported in the Results, the authors observed four salient themes in participants’ personal definitions of successful aging: 1) good health & accepting physical changes; 2) active lifestyle; 3) children’s wellbeing; and 4) independence. In the Discussion/Conclusion, the authors asserted that the evidence supports some unique views of successful aging for this Latina population relative to other populations, for example the importance of children doing well in life.

I do not have any major comments for improvement. The paper was well-organized and contained all necessary information to be a valuable contribution to the successful aging literature. I have only some minor comments below.

MINOR COMMENTS:

1. Abstract: In the Abstract Introduction paragraph, there appears to be an incomplete sentence with a question mark in the middle.

2. Results/Discussion/Conclusion: The statement of the four identified themes appeared at least three times – first paragraph of Results, first paragraph of Discussion, last paragraph of Discussion (Conclusion). As I reader, I felt this was redundant. Consider stating the list of four themes only once (in the Results), and then omit the list from the Discussion paragraphs but rather use the Discussion paragraphs to explore implications of the four themes.

6. PLOS authors have the option to publish the peer review history of their article (what does this mean?). If published, this will include your full peer review and any attached files.

Reviewer #1: No

Reviewer #2: No

Reviewer #3: **Yes: **Evan L. Thacker, PhD

---

## [Author Response · Author response to Decision Letter 0]

3 Mar 2023

Please see my response to reviewers in the file attached. Many thanks!

---

## [Decision Letter · Decision Letter 1]

6 Apr 2023

PONE-D-22-28508R1Definitions of successful aging among middle-aged Latinas residing in a rural agricultural communityPLOS ONE

Dear Dr. Ambriz,

Thank you for submitting your manuscript to PLOS ONE. After careful consideration, we feel that it has merit but does not fully meet PLOS ONE’s publication criteria as it currently stands. Therefore, we invite you to submit a revised version of the manuscript that addresses the points raised during the review process.

We look forward to receiving your revised manuscript.

Kind regards,

Frank Kyei-Arthur, Ph.D.

Academic Editor

PLOS ONE

Journal Requirements:

Additional Editor Comments:

I suggest the authors adequately address the comments of the reviewers to enhance their manuscript.

Reviewers' comments:

Reviewer's Responses to Questions

**Comments to the Author**

1. If the authors have adequately addressed your comments raised in a previous round of review and you feel that this manuscript is now acceptable for publication, you may indicate that here to bypass the “Comments to the Author” section, enter your conflict of interest statement in the “Confidential to Editor” section, and submit your "Accept" recommendation.

Reviewer #2: All comments have been addressed

Reviewer #4: (No Response)

2. Is the manuscript technically sound, and do the data support the conclusions?

Reviewer #2: Yes

Reviewer #4: Yes

3. Has the statistical analysis been performed appropriately and rigorously? 

Reviewer #2: Yes

Reviewer #4: Yes

4. Have the authors made all data underlying the findings in their manuscript fully available?

Reviewer #2: No

Reviewer #4: No

5. Is the manuscript presented in an intelligible fashion and written in standard English?

Reviewer #2: Yes

Reviewer #4: Yes

6. Review Comments to the Author

Reviewer #2: As stated to the editor, i have no further comments for the authors. Not sure why there is a minimum word count for this box.

Reviewer #4: Please see attached file.

7. PLOS authors have the option to publish the peer review history of their article (what does this mean?). If published, this will include your full peer review and any attached files.

Reviewer #2: No

Reviewer #4: No

While revising your submission, please upload your figure files to the Preflight Analysis and Conversion Engine (PACE) digital diagnostic tool, https://pacev2.apexcovantage.com/. PACE helps ensure that figures meet PLOS requirements. To use PACE, you must first register as a user. Registration is free. Then, login and navigate to the UPLOAD tab, where you will find detailed instructions on how to use the tool. If you encounter any issues or have any questions when using PACE, please email PLOS at figures@plos.org. Please note that Supporting Information files do not need this step.<quillbot-extension-portal></quillbot-extension-portal>

---

## [Author Response · Author response to Decision Letter 1]

1 May 2023

We appreciate this comment and have explicitly addressed the connection between individual/group perceptions of wellbeing/successful aging and the underlying biology of health and disease in the discussion section. This recognition strengthens our paper and the complexity of the topic of successful aging.

---

## [Editor Report · Decision Letter 2]

14 Jun 2023

PONE-D-22-28508R2Definitions of successful aging among middle-aged Latinas residing in a rural agricultural communityPLOS ONE

Dear Dr. Ambriz,

Thank you for submitting your manuscript to PLOS ONE. After careful consideration, we feel that it has merit but does not fully meet PLOS ONE’s publication criteria as it currently stands. Therefore, we invite you to submit a revised version of the manuscript that addresses the points raised during the review process.

 Thank you for addressing the previous comments of reviewers. However, you still need to address the comments raised in the previous round relating to individual/group perceptions of successful aging and the underlying biology of health and disease. Addressing this comment will strengthen your manuscript. A copy of the previous comments is provided below and is included as an attachment.

We look forward to receiving your revised manuscript.

Kind regards,

Frank Kyei-Arthur, Ph.D.

Academic Editor

PLOS ONE

Journal Requirements:

**Additional Editor Comments:**

The authors need to address the reviewer comment on the interconnections between individual/group perceptions of successful aging and the underlying biology of health and disease.

Reviewers' comments:

Ambritz and colleagues present a qualitative study on ‘successful aging’ from the perspective of

Latina women residing in a rural community. The group under study demographically

corresponds to 40 Latina women from a rural agricultural community, ethnically predominant

Mexican, within a middle age range of 40-49 (60%) and 50-59 (35%), 70% of them employed and

30% unemployed. The goal of the study was to investigate the perspectives on ‘successful aging’

by this group of Spanish speaking women, methodologically conducted by a facilitator in focus

groups, organized by age, using as a guide a questionnaire of 7 major questions about ‘successful

aging’, and contextual variations around each one of them, aiming at developing a more

complete picture of the viewpoints and perceptions on the topic by the group under

investigation. The conversations, lasting 2hs in each focus group were recorded and later

transcribed.

Good health and acceptance of body changes associated with aging, active lifestyle, children

wellbeing and independence were the major components of successful aging, as perceived by

the women group investigated.

Major comment

The premise of the study of highlighting the role played by psycho-social-cultural factors involved

in the perceptions of ‘successful aging’, and its findings as potential tools to promote and guide

policies leading to achieve healthy aging, are valuable complements of the biological perspective

on aging.

Although the present study raises an important point, for instance, the impact on the quality of

life of the participating women exerted by their children wellbeing, it is also true that if something

aggravating happens to a child it will be followed by an impact on mothers’ biology, e.g., immune

system, making them more prone to disease and degradation of life quality. The lack of explicit

recognition of the tight interconnections between individual/group perceptions of successful

aging and the underlying biology of health and disease is what this paper is missing. This

recognition strengthen rather than weakening the premise of the study while recognizing the

complexity of the topic, i.e., ‘successful aging’, that this paper approaches. This is why this

Reviewer notes that the approach utilized in this study is a ‘valuable complement’ of the

biological perspective on aging.

In regard of this, a recent investigation on cognitive decline and dementia in the Multi-Ethnic

Study of Atherosclerosis (MESA) comprising 1036 participants of Black, Chinese-American,

Hispanic and White origin, of which 53% were women, showed that it is not race/ethnicity but

cardiovascular risk factors such as age, smoking, hypertension, diabetes and socio-economic

status, that determine the outcome on cognitive decline (Austin, T.R. et al., 2022. JAHA).

<quillbot-extension-portal></quillbot-extension-portal>

---

## [Author Response · Author response to Decision Letter 2]

11 Oct 2023

We appreciate this comment and have explicitly addressed the connection between individual/group perceptions of wellbeing/successful aging and the underlying biology of health and disease in the discussion section. We have also highlighted existing research that highlights the connection between biopsychosocial factors and health. This recognition strengthens our paper and our depth of understanding of the complexity of the topic of successful aging.

---

## [Decision Letter · Decision Letter 3]

13 Nov 2023

Definitions of successful aging among middle-aged Latinas residing in a rural agricultural community

PONE-D-22-28508R3

Dear Dr. Ambriz,

We’re pleased to inform you that your manuscript has been judged scientifically suitable for publication and will be formally accepted for publication once it meets all outstanding technical requirements.

Kind regards,

Frank Kyei-Arthur, Ph.D.

Academic Editor

PLOS ONE

Additional Editor Comments (optional):

Reviewers' comments:

Reviewer's Responses to Questions

**Comments to the Author**

1. If the authors have adequately addressed your comments raised in a previous round of review and you feel that this manuscript is now acceptable for publication, you may indicate that here to bypass the “Comments to the Author” section, enter your conflict of interest statement in the “Confidential to Editor” section, and submit your "Accept" recommendation.

Reviewer #2: All comments have been addressed

Reviewer #4: (No Response)

2. Is the manuscript technically sound, and do the data support the conclusions?

Reviewer #2: Yes

Reviewer #4: (No Response)

3. Has the statistical analysis been performed appropriately and rigorously? 

Reviewer #2: N/A

Reviewer #4: (No Response)

4. Have the authors made all data underlying the findings in their manuscript fully available?

Reviewer #2: (No Response)

Reviewer #4: (No Response)

5. Is the manuscript presented in an intelligible fashion and written in standard English?

Reviewer #2: Yes

Reviewer #4: (No Response)

6. Review Comments to the Author

Reviewer #2: No further comments

No further comments

No further comments

No further comments

No further comments

No further comments

Reviewer #4: (No Response)

7. PLOS authors have the option to publish the peer review history of their article (what does this mean?). If published, this will include your full peer review and any attached files.

Reviewer #2: No

Reviewer #4: No

---

## [Editor Report · Acceptance letter]

20 Nov 2023

PONE-D-22-28508R3 

Definitions of successful aging among middle-aged Latinas residing in a rural agricultural community 

Dear Dr. Ambriz:

I'm pleased to inform you that your manuscript has been deemed suitable for publication in PLOS ONE. Congratulations! Your manuscript is now with our production department. 

Kind regards, 

on behalf of

Dr. Frank Kyei-Arthur 

Academic Editor

PLOS ONE